# Research Progress on Fatigue Life of Rubber Materials

**DOI:** 10.3390/polym14214592

**Published:** 2022-10-28

**Authors:** Xingwen Qiu, Haishan Yin, Qicheng Xing

**Affiliations:** College of Electromechanical and Engineering, Qingdao University of Science and Technology, Qingdao 266100, China

**Keywords:** rubber, fatigue, research methods, rubber formulation, environmental factors, microscopic mechanism

## Abstract

Rubber products will be fatigued when subjected to alternating loads, and working in harsh environments will worsen the fatigue performance, which will directly affect the service life of such products. Environmental factors have a great influence on rubber materials, including temperature, humidity, ozone, etc., all of which will affect rubber’s properties and among which temperature is the most important. Different rubber materials have different sensitivity to the environment, and at the same time, their own structures are different, and their bonding degree with fillers is also different, so their fatigue lives are also different. Therefore, there are generally two methods to study the fatigue life of rubber materials, namely the crack initiation method and the crack propagation method. In this paper, the research status of rubber fatigue is summarized from three aspects: research methods of rubber fatigue, factors affecting fatigue life and crack section. The effects of mechanical conditions, rubber composition and environmental factors on rubber fatigue are expounded in detail. The section of rubber fatigue cracking is expounded from macroscopic and microscopic perspectives, and a future development direction is given in order to provide reference for the research and analysis of rubber fatigue and rubber service life maximization.

## 1. Introduction

The fatigue of rubber refers to the performance deterioration and even damaging of rubber material when it is subjected to alternating loads and deformation for a long time [1]. The fatigue of rubber materials needs a time course. First, the area of a rubber material with no visible crack bears the load, and thus tiny, microscopic defects appear. Then, microscopic defects continue to expand and aggregate to form macroscopic cracks. Accordingly, there are two methods to study the fatigue properties of rubber: the crack initiation method and the crack propagation method [2]. For the crack initiation method, the decrease of material rigidity, the appearance of cracks or the appearance of complete fractures is defined as the end of life [3,4,5,6], which is based on continuum mechanics. When the initial position of a crack is determined, the crack propagation method can be used, and the crack propagation rate can be estimated by calculating the tearing energy of the material, so as to obtain the fatigue life [7].

Compared with other materials, rubber has its particularities in performance, and superelasticity and viscoelasticity are the typical characteristics of rubber. Hyperelasticity is caused by the conformational entropy change of rubber macromolecular chains, while viscoelasticity is caused by the internal friction among the components of the material. This internal friction is caused by the physical entanglement and attraction between rubber macromolecular chains, chemical cross-linking networks, filler network structures and physical adsorption and chemical adsorption on the interface, as shown in Figure 1. Superelasticity makes the stress–strain curve of rubber have nonlinear characteristics, while viscoelasticity makes the cyclic loading curve have hysteresis loops. Viscoelasticity causes part of the energy to be stored when rubber is under alternating loads, that is, elastic energy storage, while the other part is dissipated, that is, viscous energy dissipation. Elastic energy storage makes the micro-defects in rubber expand continuously, resulting in fatigue failure. The dissipated energy becomes heat, which increases the temperature of the rubber, which in turn affects the fatigue performance [8]. The characteristics of rubber such as superelasticity, viscoelasticity and temperature dependence make its research more complicated.

Rubber is a polymer material, specifically an organic polymer material. Organic polymer materials can be divided into crystalline and amorphous materials. Rubber is an amorphous material, and it is in a highly elastic state at room temperature. The main movement unit of its molecule is the “chain segment”. In recent years, many scholars have used molecular dynamics to study the fatigue life of rubber materials, and they think that the damaging of rubber is the breaking of chemical bonds in the main chain or the damaging of the interaction force between polymer chains (i.e., the secondary bond). From the microstructure point of view, a crack is the formation of a new surface, and there are two ways to form a new surface: in the first, the stress causes the chemical bond to break, thus causing the molecular chain to break and forming a crack; in the second, shear causes the destruction of secondary bonds, which leads to relative slippage of molecular chains and cracks. Therefore, the “chain segment” is the main movement unit of rubber material, and its length and shape will affect the performance of rubber [9].

The references in this paper come from two resource libraries: Web of Science and Scopus Data. Using “rubber”, ”fatigue” and “life” as keywords to search, a total of about 118 articles published in recent 20 years was obtained, as shown in Figure 2.

To sum up, the fatigue performance of rubber materials is affected by many aspects, including filler, load, environmental factors, etc. In this paper, the research status of rubber fatigue will be expounded from both micro and macro aspects. The research on rubber fatigue will be described from the following aspects: the second chapter is the research of specimens, the third chapter is the discussion of fatigue research methods, the fourth chapter discusses the factors that affect the fatigue life of rubber, the fifth chapter is the research of rubber section, and the sixth chapter is the expectations and conclusions. As shown in Table 1, we have summarized the important literature of each chapter so that readers can better understand the structure of this article.

## 2. Fatigue Specimens

According to the fatigue research methods of rubber materials, different rubber specimens are used, as shown in Figure 3. Figure 3a represents dumbbell-shaped specimens, whose test procedures and geometric shapes are formulated according to ASTM standards and ISO standards, and Figure 3b represents diabolo-shaped specimens, all of which are used for crack initiation tests. Among them, diabolo-shaped specimens are also called dumbbell-shaped cylindrical or hourglass-shaped, and their specific sizes are not uniform because there is no standard for fatigue tests. Diabolo specimens are often used for fatigue research in laboratories because they can be subjected to combined fatigue tests of tension, compression and torsion. In 1940, Cadwell et al. [10] used diabolo specimens for rubber fatigue tests; later, some scholars used this shape of specimen, but the sizes were different. The two most common sizes in the literature were proposed by Beatty [11]. In first one, the curvature radius of the middle part is 42 mm, as shown in Figure 3b, while in the second one it is 2 mm, as shown in Figure 3c. This kind of diabolo specimen can ensure that stress and strain are concentrated in the middle part, so that cracks appear in the middle part. In addition, the middle part in Figure 3c is thinner, and the torque required for applying torsional load is smaller, so it is suitable for fatigue testing with torsion. For the convenience of experiments, diabolo-shaped specimens are usually vulcanized together, with metal plates at both ends during vulcanization. However, this structure also makes it easy to cause cracks or even fractures at the interface between the metal plate and the rubber during the experiment. LeCam et al. [12] explained that this phenomenon is due to cavitation between the metal plate and the rubber specimen. Figure 3d shows the torn specimen used in a crack propagation experiment. Before the experiment, a crack with a specific length is preset on the specimen because this can make the crack propagation proceed along the preset crack and prevent multiple scattered cracks or crack deviation [13]. In fact, the shape of the specimen is very important for rubber fatigue tests because the shape of the specimen appearing in the current standards and literature cannot reflect the multiaxial load of real rubber products when they work.

## 3. Rubber Fatigue Research Methods

The crack propagation process of rubber can be divided into two stages: first, there is no obvious crack in the rubber, and microscopic defects begin to sprout and continuously gather to form cracks; in the second stage, the tiny cracks formed in the previous stage continue to expand until local damage or even the whole fracture fails [14]. According to these two stages, the research methods of rubber fatigue are divided into the crack initiation method and the crack propagation method by relevant scholars. With the continuous deepening of continuum mechanics and the research on micro-mechanisms of fractures, Gent [15] first pointed out that the internal mechanism of micro-defect initiation and propagation is consistent, that is, the two processes of crack initiation and crack propagation should be studied in a unified way. More and more scholars agree with this view and have conducted much research on the topic [8,16].

When calculating the life of rubber materials by the above two research methods of rubber fatigue, it is necessary to define the conditions of life termination. However, there is no definite and unified standard to define the life of such materials, and three commonly used life termination conditions can be found in the literature. The first is that the rubber specimen breaks as the end of its life [17,18,19,20]. Considering the experimental time required for the specimen to break, a rubber specimen with a smaller volume is suitable for this situation. The second involves cracks of a certain size appearing. Wang et al. [21] and Belkhiria et al. [22] regard the appearance of a 1 mm visible crack as the end of life. Because the crack length is difficult to observe during experiments, some scholars also regard the load drop as the criterion of the end of life. Woo et al. [23] regard a maximum load drop of 20% as the criterion of fatigue failure, and Lemire et al. [24] considers a reduction of the maximum axial force by 50% as the relevant criterion. The third is a decrease of effective stiffness as the criterion of fatigue failure. Effective stiffness is defined as the ratio of the maximum load to the maximum displacement in a cycle, which is suitable for fatigue experiments of load control or displacement control. Seichter et al. [25] compared the three criteria and finally chose the decrease of effective stiffness as the judgment condition of the end of life. Champy et al. [26] also took the decrease of effective stiffness as the fatigue criterion.

### 3.1. Crack Initiation Method

The crack initiation method was first put forward by the German engineer Wöhler during a fatigue experiment using railway axle components. He used the number of cycles as a metric to evaluate the fatigue life and drew the S–N curve (i.e., stress–strain curve), which is still in use today. Cadwell [10] first used it to evaluate the fatigue life of rubber. Shangguan et al. [27] found that the crack initiation life accounted for more than 90% of the total life of rubber through a uniaxial tensile fatigue test of filled natural rubber, so it can be approximately considered that the crack initiation life is the fatigue life of rubber.

There are two preconditions for applying the crack initiation method: first, the rubber material is isotropic; second, the rubber material has no macroscopic cracks. The crack initiation method is a research method based on continuum mechanics and damage mechanics. It is considered that the fatigue life of a material has a power series relationship with its mechanical parameters (also called fatigue damage parameters and fatigue prediction factors). The commonly used damage parameters are stress, strain and strain energy density. Damage evolution is also a commonly used fatigue life criterion.

#### 3.1.1. Strain-Based Life Prediction

Strain is a commonly used damage parameter in fatigue life calculation because it can be easily obtained from displacement [28]. The maximum principal strain is widely used because the crack propagation direction is always perpendicular to the maximum principal strain direction [29]. Li et al. [22] set up a life prediction model based on a uniaxial tensile fatigue test with the maximum principal strain as the fatigue damage parameter, and the prediction results are in good agreement with the experimental results. Shangguan et al. [27] carried out uniaxial tensile fatigue tests on dumbbell-shaped specimens and diabolo-shaped specimens of carbon-filled natural rubber and compared the fatigue life models established with principal strain, octahedral shear strain and strain energy density as damage parameters. It was found that the correlation between the predicted results and the experimental results was good, with the Green–Lagrange strain peak value performing best as a damage parameter. Belkhiria et al. [30] established fatigue life prediction models of rubber materials based on logarithmic strain, engineering strain, Green–Lagrange strain, Euler strain and octahedral shear strain and verified the accuracy of several models through the uniaxial tensile fatigue testing of styrene–butadiene rubber. It was concluded that the selected strain parameters could describe the fatigue life well, and the model based on octahedral shear strain was the most accurate. Yin Fang et al. [31] reached a similar conclusion. Compared with [27,30], we can conclude that life calculation models with different damage parameters are suitable for different kinds of rubber.

The results obtained by the above scholars are all based on uniaxial fatigue tests, and the conclusions are all very good. However, uniaxial fatigue tests cannot represent the actual load of rubber products. Ayoub et al. [32] compared the ability of several different damage parameters to predict the multiaxial fatigue life of rubber. Multiaxial fatigue testing includes combined experiments on tension–compression and torsion, and they found that the maximum principal strain cannot unify the uniaxial and multiaxial fatigue life well. Lai Yaling [33] also came to the same conclusion in her paper.

#### 3.1.2. Stress-Based Life Prediction

Stress is also a commonly used parameter to study the fatigue life of rubber. Saintier et al. [34] used the first and second invariants of Cauchy stress as fatigue criteria to predict the multiaxial fatigue life of natural rubber and found that the multiaxial fatigue life of rubber could not be predicted well based on stress criteria. Poisson et al. [35] compared uniaxial and multiaxial fatigue tests of chloroprene rubber and found that the first principal stress could not predict its fatigue life well. They analyzed the fatigue failure surface with scanning electron microscopy and observed that the tongue on the failure surface of the specimen was subjected to uniaxial fatigue with a high R ratio, but no tongue was found on any multiaxial fatigue failure surface. They concluded that this might be the result of the crystallization of chloroprene rubber under uniaxial tension.

With the development of continuum mechanics and damage mechanics, Wang et al. [36] established a calculation model for predicting the uniaxial tensile fatigue life of carbon-filled natural rubber based on continuous damage mechanics and obtained good prediction results; Ayoub et al. [37,38] extended the continuous damage mechanics model to the multiaxial fatigue of rubber based on Wang’s work, and they combined the damage mechanics and the cracking energy density proposed by Mars [39] to derive effective stress parameters, which were used in multiaxial fatigue life calculation and achieved good results.

#### 3.1.3. Life Prediction Based on Energy

Strain energy density is also a relatively simple fatigue damage parameter. Pan et al. [40] predicted the fatigue life of ceria-filled natural rubber with strain energy density as a damage parameter and found that the prediction results were not very good because all energy could not be released for crack propagation, so the strain energy density parameter was not suitable for predicting the multiaxial fatigue life. Ayoub et al. [32] also pointed out that the strain energy density cannot be a qualified fatigue parameter when predicting the fatigue life of styrene–butadiene rubber, and because the strain energy density is scalar, it only depends on the final state of the load, so it cannot predict the direction of crack generation.

Mars [39] created a new prediction parameter called crack energy density, which extracted the part of strain energy density used for crack propagation. He defined cracking energy density as the stored elastic energy density, which can be used for cracks on a given material plane and can be calculated for any complicated strain history. The theory of cracking energy density has two key points: the critical plane and the energy release rate. Zine et al. [41] compared the strain energy density and cracking energy density to predict the multiaxial fatigue life of carbon-filled styrene–butadiene rubber and found that the cracking energy density standard was indeed better than the strain energy density; Belkhiria et al. [42] and Pan [40] reached the same conclusion. Poisson et al. [43] studied the multiaxial fatigue behavior of polychloroprene rubber based on the dissipative energy density criterion and compared the predictive ability of the dissipative energy density criterion with the strain energy density and the first principal stress regarding multiaxial fatigue. The dissipative energy density criterion obtained satisfactory results. Marco et al. [44] think that the ratio between the total energy dissipated and the energy related to the fatigue mechanism of rubber depends on the crack surface density.

Tobajas et al. [45] proposed a new fatigue prediction method, which considers stress, strain and strain energy at the same time and combines them into a new fatigue parameter according to the proportion of its influence on fatigue life. It has been proven that the new parameter is reliable for fatigue life prediction by comparison with other parameters.

#### 3.1.4. Miner Linear Damage Criterion

The Miner linear damage criterion is a cumulative damage model. When the number of applied cycles divided by the number of failed cycles equals 1, this is defined as fatigue failure. Harbour et al. [46] studied the multiaxial fatigue life of natural rubber and styrene–butadiene rubber and predicted the fatigue life of natural rubber by using Miner’s linear criterion combined with cracking energy density. However, the applicability of this criterion to styrene–butadiene rubber was poor, and they attributed this to a large number of cracks in the natural rubber. Chen et al. [47] said that the Miner criterion can unify constant amplitude fatigue calculation and variable amplitude fatigue calculation when studying the fatigue life of rubber vibration isolators. Zarrin-Ghalami et al. [48] used the Miner criterion to predict the fatigue life of natural rubber under variable amplitude loads and achieved good results.

### 3.2. Crack Propagation Method

The crack propagation method is a research method based on fracture mechanics. It presumes that there will inevitably be tiny cracks in rubber products due to the production conditions, production technology, material itself and other reasons. Fatigue life is defined as the number of cycles required for a crack with a given initial size (usually 1 mm) to expand to the maximum crack size under the action of the external energy release rate [49]. Therefore, it is usually necessary to preset a crack on the specimen when applying the crack propagation method. Mars [50] said that compared with the crack initiation method, the crack propagation method can avoid unexpected cracks, and only a few experiments are needed to characterize the whole range of fatigue behavior.

There are two important factors in the crack propagation method: the energy release rate and the crack propagation rate. For rubber materials, the potential energy inside the material is usually released reversibly or irreversibly, which is defined as the change of potential energy stored in rubber materials for each additional unit crack, also known as the tearing energy. Lake and Lindley et al. [51,52] conducted crack propagation experiments on natural rubber and styrene–butadiene rubber. According to the characteristics of crack propagation, the crack propagation process was plotted and divided into four stages, and a fatigue life model was established for each stage. Asare et al. [53] verified the fatigue crack propagation behavior of rubber at room temperature and high temperature based on fracture mechanics and proved the effectiveness of fracture mechanics.

Shangguan Wenbin and Wang Xiaoli et al. [54] carried out crack propagation tests on torn rubber specimens with single notches under variable amplitude loads and established a crack propagation calculation model. Ghosh et al. [55] studied the crack propagation behavior of blends composed of natural rubber and polybutadiene rubber and found that the crack propagation rate increased rapidly beyond a certain strain level, which appeared between 20% and 35% strain, and varied with the amount of polybutadiene rubber. It was also found that 60/40 blends showed a faster crack growth rate due to higher heat production. Pei Shuo et al. [56] deduced the functional relationship between tearing energy and crack growth rate based on fracture mechanics. Ait-Bachir et al. [57] showed that the energy release rate of a small crack is always proportional to the crack size and that it has nothing to do with the load condition or crack direction. Fukahori et al. [58] explained the transition of the relationship between the critical strain energy release rate and the critical crack growth rate according to the phenomenon of elastic viscosity transition. The new elastic–viscous transition diagram consists of three regions: the elastic brittle fracture zone, the viscoelastic fracture zone and the intermediate transition zone between the elastic and viscous zones. The transition zone characterized by stick–slip motion is the result of unstable fluctuation of the crack propagation rate due to energy dissipation near the glass transition temperature. Meanwhile, Netzker et al. [59] analyzed the fracture behavior of rubber materials based on the global energy balance. They found that the energy loss is due to the increase of the dissipation zone, rather than the limited stable crack propagation zone.

## 4. Factors Affecting the Fatigue Life of Rubber

In order to prevent the failure of rubber materials, it is very important to know the factors that affect their fatigue life. Mars [60] made an extensive summary of the factors affecting the fatigue life of rubber in 2004. This paper will discuss three aspects: mechanical load, rubber composition and environmental factors. Most articles discuss the influence of mechanical load on rubber fatigue, including stress and strain amplitude, frequency and loading mode. The fatigue life of rubber is not only related to the type of rubber but also to the filler. Environmental factors affecting rubber fatigue include temperature, ozone, humidity, etc.

### 4.1. Mechanical Conditions

The fatigue life of rubber is closely related to its mechanical load. Mechanical loads include tension–tension, tension–compression, compression–compression, tension–torsion and compression–torsion loads [26].

In fatigue tests, the influence of different loading modes on fatigue life can be studied by applying constant amplitude or variable amplitude load. Abraham et al. [61] studied the effect of strain amplitude on the fatigue life of styrene–butadiene rubber and ethylene–propylene rubber and found that when the strain amplitude is constant, increasing the minimum strain can improve their fatigue life. Many researchers pay attention to the constant strain amplitude when studying rubber fatigue [62,63], while Harbour et al. [46] proposed that the fatigue life should be studied under variable amplitude load and multi-axial load because this is more in line with the complex loads that rubber materials bear in service. Gehrmann et al. [64] claimed that controlled displacement experiments were not equal to the controlled strain because even if the displacement amplitude was constant, the strain of the narrow part would change significantly, so a method was proposed to transform the changed strain into a constant equivalent strain which conformed to the Wöhler curve.

### 4.2. Rubber Composition

As shown in Figure 4, natural rubber accounts for nearly half of the rubber types studied in recent years. Styrene–butadiene rubber is the most studied synthetic rubber. Different rubbers are widely used in the automobile industry because of their specific properties; for example, hydrogenated nitrile rubber has good oil resistance and heat resistance [65]. Magnetorheological elastomers can be used in shock absorbers, vibration isolators, etc., and have great potential [66].

An important feature of natural rubber is crystallization, which makes it different from synthetic rubber. Many scholars have studied the effect of natural rubber crystallization on fatigue life [67,68,69]. Many factors can lead to the crystallization of natural rubber, such as temperature and strain, that is, low-temperature crystallization and strain-induced crystallization. Strain-induced crystallization can greatly improve the fatigue life of rubber materials [70] because the generated microcrystal structure makes the originally disordered rubber molecular chains orient along the stress direction, thus making the local rubber molecular chains arrange regularly, resulting in stronger fatigue resistance [71]. Champy et al. [26] established a Haigh diagram of a wide range of positive load ratios based on tensile–tensile load testing. These results emphasized the important role of strain-induced crystallization in the fatigue life of crystalline rubber materials. Sridharan et al. [72] compared the mechanical properties of natural rubber and polybutadiene rubber and found that polybutadiene rubber was more sensitive to temperature.

Different kinds of rubber have different mechanical properties, and fillers also have an important influence on the mechanical properties of rubber materials. Carbon black is the most commonly used reinforcing material, and sometimes white carbon black is added to enhance the fatigue resistance or tear resistance. Commonly used carbon blacks are N234, N115, N550, etc., and one or more of them can be used together to achieve good mechanical properties of rubber materials [73]. At this time, many researchers are still studying the optimal dosage of carbon black [74,75,76]. Dinari et al. [77] studied the uniaxial tensile fatigue behavior of SBR filled with carbon black and found that the interface interaction between carbon black and SBR increased with the increase of carbon black content.

The size of the filler is also a key factor affecting the fracture and fatigue of rubber, and many scholars are also exploring the influence of different fillers on rubber properties. Dong et al. [78] studied the effects of zero-dimensional spherical carbon black, one-dimensional fibrous carbon nanotubes and two-dimensional planar graphene oxide on the mechanical properties of rubber. By adding different parts, the mechanical properties were studied at the same hardness. It was found that carbon black-filled natural rubber had the best tear resistance and fatigue resistance, while planar graphene had a limited anti-fatigue effect, and carbon nanotube-filled natural rubber exhibited weakened anti-fatigue performance due to high hysteresis loss. Xu et al. [79] studied the influence of carbon nanotubes and graphene oxide on the microstructure and fatigue properties of silica/styrene–butadiene rubber composites and found that the synergistic effect of silica and carbon nanotubes or graphene oxide was beneficial to the formation of filler networks, improved the mechanical properties of styrene–butadiene rubber and reduced the crack propagation rate, and the tip of the crack propagation was easy to deflect. Chen et al. [80] studied the reinforcing effect of silica nanoparticles on silicone rubber and found that its ultimate strength and toughness increased monotonically with the increase of the mass fraction of nanoparticles, which was attributed to the blocking effect of nano-sized aggregates on crack propagation and the weak interfacial debonding between aggregates and the matrix. Yao et al. [81] added a silane coupling agent to natural rubber filled with silica and found that the silane coupling agent could promote the dispersion of silica. Scanning electron microscopy showed that there were uniformly distributed tiny ligaments and dimples at the crack tip, which could weaken the crack propagation. Duan et al. [82,83] carried out uniaxial fatigue tests on vulcanized natural rubber filled with and without cerium oxide. By analyzing the microstructure of the fracture of the sample with a scanning electron microscope, it was found that cerium oxide had good dispersibility, which improved the internal network structure of the rubber, enhanced the interaction of rubber molecular chains and improved the anti-fatigue performance of the rubber.

### 4.3. Environmental Factors

In addition to the mechanical load and rubber composition, environmental factors are also important factors that affect the fatigue life of rubber [84], and the fatigue life of rubber will be quite different in different environments.

Most rubber products are used in open air, so most of the research on rubber fatigue is in the air medium. LeGorju et al. [85] found that oxygen in the air can make rubber materials undergo chemical reaction; especially at high temperatures, the oxidation reaction is more intense. Temperature is a crucial factor affecting the fatigue life of rubber. Different kinds of rubber materials have different sensitivities to temperature. When the ambient temperature is raised from 0 to 110 °C, the fatigue life of natural rubber is shortened by 4 times, while that of styrene–butadiene rubber is shortened by 10,000 times [86]. Ruellan et al. [87] carried out uniaxial tensile fatigue tests on carbon-filled natural rubber at different temperatures and found that strain-induced crystallization was obviously observed on the fracture surface at room temperature, but at 90 °C, the crystallization was obviously reduced and completely disappeared at 110 °C, so high temperature affects the crystallization of natural rubber. Ngolemasango et al. [88] also observed this phenomenon in their experiment. Rey et al. [89] studied the effect of temperature on carbon-filled and unfilled silicone rubber. When the microstructure is stable, the hardness of unfilled silicone rubber increases with the increase of temperature. For carbon-filled silicone rubber, hysteresis, stress relaxation and stress softening decrease with the increase of temperature. Haroonabadi et al. [90] thermally aged acrylonitrile–butadiene rubber (NBR) for 7 days and found that its crosslinking density increased, while its tensile and tear strength decreased. Chou et al. [91] found that the life of EPDM rubber filled or unfilled with carbon black decreased after thermal aging for 6 months. As the thermal oxygen reaction degrades and reforms the network, increasing the aging temperature and aging time will irreversibly reduce the fatigue life of rubber [92]. Luo et al. [93] carried out fatigue tests on hourglass rubber specimens. The experiments showed that the surface temperature of the specimens kept a stable value for a long time, which accounted for most of the fatigue life, and then increased sharply until the specimens cracked. The sharp rise of temperature can be regarded as a precursor of fatigue failure. Therefore, the relationship between steady-state temperature rise and maximum principal strain is established, and the fatigue life of rubber is determined.

At the same time, ozone will react chemically with rubber materials, leading to their accelerated damaging. Vinod et al. [94] found that adding aluminum powder can improve the ozone resistance of natural rubber. Although the samples exposed to ozone also produced cracks, the cracks were smaller and less continuous than those of natural rubber without aluminum powder. Zheng et al. [95] chose squalene, a small molecular substance with a similar structure to that of natural rubber, to simulate the ozone aging process of natural rubber. It was found that the molecular network of natural rubber was destroyed during the aging process, and three new free radicals (H, O_2−_ and C = C) were produced, which led to the weakening of the rubber’s mechanical properties. Kamaruddin et al. [96] studied crack propagation under different tensile strains in an ozone environment and found that the cracks formed under low strain were few and could grow very long, while the cracks formed under high strain were many but short, which they claimed was due to the interference between ozone and tensile loading. Iwase et al. [97] studied the ozone degradation of vulcanized isoprene rubber by changing the air humidity. It was found that black powder appeared on the rubber surface when the relative humidity was greater than 50%, but no black powder appeared when the relative humidity reached 50%. The severe ozone degradation of rubber under high humidity was attributed to hydroxyl radicals generated by the reaction of water vapor and ozone. However, Kamaruddin et al. [94] claimed that rubber would resist the attack of ozone in the presence of water.

The fatigue life of rubber in seawater has also attracted the attention of scholars. Narynbek et al. [98] studied the fatigue performance of carbon-filled natural rubber in seawater through uniaxial fatigue testing and found that the results obtained under relaxed loading conditions were almost the same as those in air, while the fatigue life in seawater under non-relaxed loading conditions under large strain was longer than that in air. This phenomenon is due to the fact that the thermal conductivity of seawater is higher than that of air, which can effectively reduce the temperature of rubber. However, LeGac et al. [99] conducted a uniaxial fatigue test under the condition of non-relaxed loading and a load ratio of 0.2, finding that the fatigue life of rubber in seawater was lower than that in air, which was attributed to the decrease of the strain-induced crystallization level of rubber in seawater. When the load ratio increased, the fatigue life of rubber also increased.

## 5. Cross-Section Study

From the above chapters, it can be seen that the fatigue performance of rubber materials is affected by the loading mode and environmental factors, so their crack sections also have different forms. This chapter will elaborate from macro and micro perspectives.

### 5.1. Macro-Section Study

Under different loading modes, the ways of crack growth are quite different. Hainsworth et al. [100] carried out a crack growth experiment using silicone rubber and observed the cracking by using environmental scanning electron microscopy. It was found that the crack first appeared at the edge of the specimen and expanded along the edge instead of along the thickness. Moreover, the crack growth of silicone rubber was not a continuous fracture process because a large number of ligaments formed by local crystallization at the crack prevented the crack growth, and the crack underwent explosive growth after about 100 fatigue cycles. After 82% of the fatigue cycle, cracks also began to appear in the middle of the specimen. These cracks originated from processing defects in the specimen and propagated along the direction perpendicular to the loading direction. Lecam et al. [101] studied the fatigue failure forms of diabolo carbon black-filled rubber under three conditions: uniaxial tension–compression, relaxed uniaxial tension and non-relaxed uniaxial tension. It was found that the fatigue forms of uniaxial tension–compression and relaxed uniaxial tension were the same, and they all appeared on the surface of the middle part of the specimen, so it was concluded that compression had no effect on rubber fatigue. In the non-relaxation tensile test, a variety of crack forms were found, and crack branches appeared, which was because microcrystals formed by strain-induced crystallization hindered the crack propagation. After that, Lecam et al. [102] conducted a multiaxial fatigue test—that is, torsion was added to the tensile test—and the torsion was 0 when it reached the maximum value, and the tension was 0 when it reached the maximum value. The experimental results show that, compared with uniaxial experiment, there are only two fatigue forms in multiaxial experiments: one is that the crack starts from the surface, and the other is that the crack starts from the inside of the specimen and spreads. Xiang et al. [103] studied the crack propagation of vulcanized natural rubber and found that the crack propagation path depends on the R ratio (i.e., the ratio of the minimum load to the maximum load); when the R ratio was kept at 1/3, all samples showed the same crack propagation path.

### 5.2. Micro-Section Study

#### 5.2.1. Crack Initiation

Of course, to study the mechanism of crack initiation and propagation, it is necessary not only to observe the location and propagation direction of crack initiation macroscopically but also to study its internal mechanism microscopically. Lecam et al. [102] observed the initiation and propagation process of cracks in rubber materials by scanning electron microscopy and found that the cracks did not completely originate from the crack surface, but were 200–600 microns away from the surface. They explained that the cracks were caused by molding: due to the low thermal conductivity of rubber, molding would produce a skin effect, that is, the material stiffness of the sample surface and that of the main body were different [104]. At the same time, it was concluded that the crack initiation begins with the failure of carbon black aggregates, air pockets at the poles of carbon black aggregates, the debonding of oxide and the rubber matrix, and the concentration of a large number of holes. Granfcoin et al. [105] also observed debonding between the filler and matrix after rubber fatigue testing and concluded that debonding was the main microscopic mechanism leading to the formation of microcracks. In addition, Hainsworth et al. [98] and Granfcoin et al. [105] both found aggregates in the original samples.

Pérocheau et al. [106] used stretching and electron beams to break the cross-linking between macromolecules and found that the debonding between the filler and matrix caused the cavity at the crack tip. Glanowski et al. [107] observed the fatigue of carbon-filled natural rubber by X-ray computer micro-tomography and obtained two damage mechanisms: one is the cavitation phenomenon at the two poles of aggregates, and the other is the fracture of aggregates; Lesaux et al. [108] reached the same conclusion. Federico et al. [109] analyzed styrene–butadiene rubber filled with silica by microcomputer X-ray tomography and observed that the size of silica aggregates ranged from 6 microns to 140 microns, and the sphericity ranged from 0.35–1. It was found that large and non-elliptical silica aggregates were the source of cavity formation. However, after high temperature treatment, it was concluded that the cavity is mainly caused by the decomposition of aggregates. The authors explain that increasing the temperature increases the stiffness of the matrix, which leads to stress transfer at the interface between aggregates and the matrix. Huneau et al. [110] studied the fatigue cracking of natural rubber filled with carbon and found that the crack initiation mechanism of carbon black aggregates is different from that of oxide aggregates, and only the crack initiated by carbon black aggregates is accompanied by crack propagation because carbon black has stronger cohesion and adhesion to the matrix, and its cohesion is stronger than its adhesion. Candau et al. [111] used an experimental device based on a four-camera stereo vision system to observe the fatigue of EPDM rubber and found that the macroscopic increase of volume was due to the increase of the rubber void ratio, and in rubber with a large filler content, the volume change showed an upward trend; Ilseng et al. [112] reached the same conclusion through uniaxial tensile fatigue tests of nitrile rubber and fluororubber. Weng et al. [113] observed nano-scale voids in vulcanized natural rubber under high temperature and small strain amplitude. The analysis showed that the thermal oxidation effect led to the destruction of the cross-linked structure near ZnS particles, and the cyclic load made ZnS particles separate to produce voids.

#### 5.2.2. Crack Propagation

After the micro-defects mentioned in the previous section are generated, they will continue to expand to form micro-cracks, that is, after a crack is initiated, the crack tip will continue to expand under the action of load, eventually forming macro-cracks until the fatigue failure of the material. Therefore, the microscopic mechanism of crack initiation and propagation needs to be studied in order to better understand the fatigue process of rubber.

Beurrot et al. [114] used scanning electron microscopy to observe the crack tip in real time at the crack propagation stage of carbon-filled natural rubber and found that the crack tip was composed of rhombic areas separated by ligaments, and the crack propagation was driven by ligament ruptures, which occurred continuously in all areas of the crack front surface, but at different speeds; this could explain the initiation and limited propagation of branches deviating from the main crack. Yan et al. [115] compared the crack propagation of natural rubber filled with and without graphene and found that adding graphene slowed the crack propagation at higher strains of 40% and 50%, while at lower strains of 20% and 30%, the addition of graphene accelerated the crack propagation. The authors explained that this was related to strain-induced crystallization at the crack tip: at low strain, almost no strain-induced crystallization occurred, and adding graphene made the crack tip void. However, under high strain, microcrystals produced by strain-induced crystallization hindered crack propagation, and branch cracks produced new surfaces, which dissipated more energy for main crack propagation. Weng et al. [116] studied the morphology change of the crack tip of vulcanized natural rubber under different tearing energies, and they found that there was a turning point in the crack growth rate. When the tearing energy T < 600 J/m^2^, the surface of the crack tip appeared with surface buckling and peeling, which caused secondary cracking, so the crack propagation rate was low. However, when the tearing energy T > 600 J/m^2^, the crack propagation became driven by the ligament rupture mechanism.

## 6. Expectations and Conclusions

The calculation of rubber fatigue life is usually divided into two methods: the crack initiation method and the crack propagation method. The crack initiation method can find the power relation between fatigue parameters and life, which has good prediction accuracy for uniaxial fatigue life, but it is still difficult to unify multiaxial fatigue life. Therefore, it is of great significance to develop a new life model to unify the uniaxial and multiaxial fatigue lives of rubber materials. The difficulty of the crack propagation method lies in calculating the energy release rate.

In terms of rubber filler, carbon black is the most widely used. The addition of carbon black can improve the critical tear energy, and at the same time, carbon black is not easy to disperse, which can easily cause the aggregates to be too large and make the initial defect size larger. Therefore, researchers look for alternative fillers that can enhance the combination of the rubber matrix and filler, thus improving the fatigue performance of rubber [117]. Among them, carbon nanotubes are frequently used fillers which can provide excellent thermal, mechanical and electrical properties [118]. However, carbon nanotubes also have the problem of uneven dispersion, which is a subject to be further studied in the future.

More and more researchers have realized the influence of temperature on the fatigue performance of rubber, and high temperature will further reduce the fatigue performance of rubber [85]. Studies of the influence of temperature on rubber should include two aspects: one is the ambient temperature, and the other is the self-heating temperature caused by cyclic load. Most literature focuses on increasing the ambient temperature to study the influence of temperature on rubber, but the self-heating is often neglected. When rubber is subjected to cyclic load, it will self-heat due to its viscoelasticity. Because there is no standard to evaluate the self-heating phenomenon, more research needs to be conducted to examine the self-heating of rubber. Seichter et al. [25] studied the effect of frequency on rubber fatigue and found that frequency would not affect the fatigue life of rubber, but high frequency would lead to self-heating of rubber, which would increase the temperature of rubber and reduce its life. Therefore, collecting complete data on the influence of temperature on the fatigue of different kinds of rubber is helpful to quickly evaluating the life of rubber.

The geometry of rubber specimens also has an important influence on rubber fatigue. Dumbbell specimens in ASTM and ISO standards are widely used, but there is no uniform standard for diabolo specimens, which leads to different sizes of diabolo specimens being used by researchers. Therefore, it is necessary to establish a unified standard to conduct such experiments so that the results can be compared.

Many researchers have observed and analyzed crack sections in order to understand the microscopic mechanism of rubber fatigue. Among them, crack initiation is due to the debonding between aggregates and the rubber matrix and the fracture of aggregates, and crack propagation is driven by ligament fractures. Crack initiation and propagation are greatly affected by temperature, and the mechanism of initiation and fracture is different at different temperatures, so it is necessary to study the effect of temperature on the fatigue performance of rubber.

## Figures and Tables

**Figure 1 polymers-14-04592-f001:**
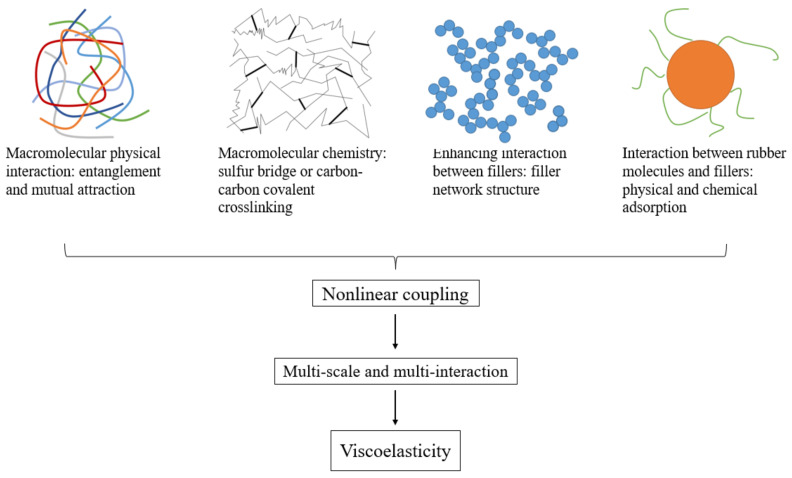
Four main interactions within rubber-based nanocomposites.

**Figure 2 polymers-14-04592-f002:**
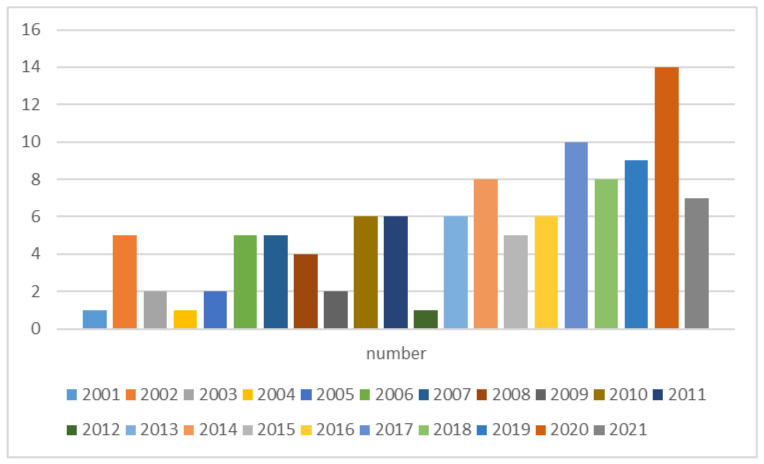
Number of literature articles in the last 20 years.

**Figure 3 polymers-14-04592-f003:**
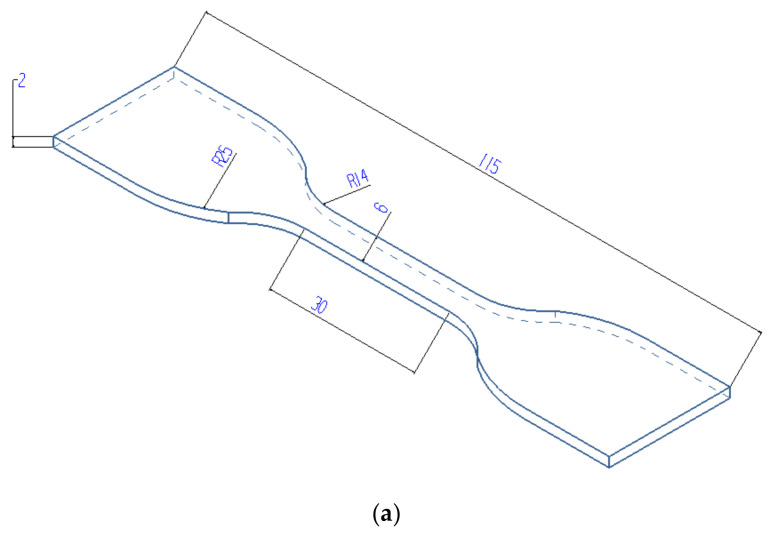
Rubber fatigue specimens: (**a**) dumbbell-shaped specimen; (**b**) diabolo-shaped specimen; (**c**) diabolo-shaped specimen; (**d**) torn specimen.

**Figure 4 polymers-14-04592-f004:**
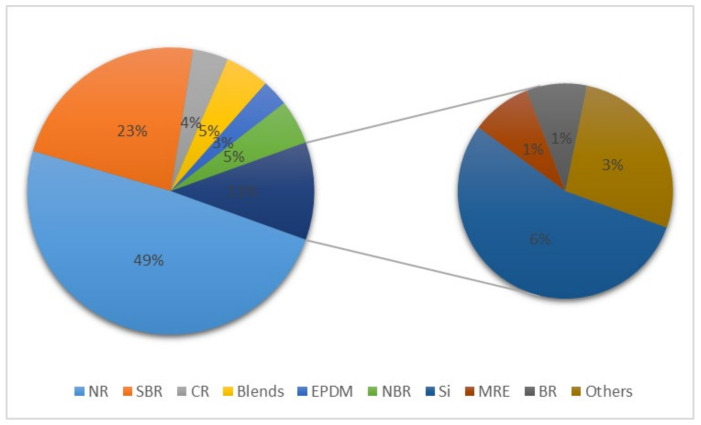
Proportion of rubber types studied in the literature.

**Table 1 polymers-14-04592-t001:** Some important references in the field of fatigue mentioned in this paper.

Chapter		Author	Contribution
2		Beatty	Propose two common dimensions of diabolo specimen.
		Gent	He introduced torn specimens for fatigue experiments.
3		Zine, Wang, Woo et al.	They define different end-of-life conditions.
	Crack initiation method	Cadwell	He introduced the crack initiation method into the calculation of rubber fatigue life.
		Ayoub	He found that taking strain as a damage parameter could not predict multiaxial fatigue life well.
		Wang, Ayoub	They established a model for calculating the fatigue life of rubber based on damage mechanics.
		Mars	He proposed a new parameter called cracking energy density, that is, the part of strain energy density used for crack propagation.
	Crack propagation method	Lake, Lindley	They divided the crack propagation process into four stages.
4	Rubber composition	Yao, Dong	They studied the influence of different fillers on the fatigue life of rubber.
	Environmental factors	Ruellan	He found that temperature can affect the crystallization of natural rubber.
5	Macro-section study	Le cam	He studied the influence of different loading modes on crack initiation and propagation.
	Micro-section study	Le cam, Granfcoin	They summarized the microscopic mechanism of crack initiation.
		Weng	He studied the mechanism promoting crack propagation.

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
