# Peer review of "Research Progress on Fatigue Life of Rubber Materials"

_polymers, 2022, doi:10.3390/polym14214592_

Round 1

Reviewer 1 Report

The references are sufficient, and the numbers of past 5 years referencing are adequate. While nicely written, I see no research gap in the paper, but more of a literature review kind of write up is portrayed in this review paper. I would suggest to add a clear research gap, and resubmit. 

Author Response

Dear reviewers,
Hello, I wrote the research deficiencies of each part in the last chapter "Expectation and Conclusions" of the manuscript. I wonder if it is sufficient? Besides, the red font in the manuscript is grammatical errors that I found and corrected.
I look forward to your reply.

Yours,

Xingwen Qiu

Reviewer 2 Report

Manuscript ID: polymers-1957730

Review Report

The reviewer would like to submit the review report to the manuscript entitled "Research progress on fatigue life of rubber materials," which might be considered for publication in the Polymers (ISSN 2073-4360).

Following the completion of the review process, the reviewer would like to provide some critical thoughts and ideas to assist the authors in working more effectively.

Comments and Suggestions for Authors

The authors presented a review topic regarding rubber materials' fatigue life, which interested readers. Rubber products fatigue under alternating loads, and their fatigue performance deteriorates in hostile environments, reducing their service life. This paper summarizes the current research status of rubber fatigue from three aspects: research methods, factors affecting fatigue life, and crack section, including a comparison of different rubber specimens, a summary of two research methods, and the effects of mechanical conditions, rubber composition, and environmental factors. Macroscopic and microscopic views of rubber fatigue crack are offered, along with future development directions to aid with rubber fatigue research and life extension. The reviewer concluded that the paper has potential and could be published. Please revise based on the reviewer's comments.

General comments

In general, this manuscript is well-written, and it contributes to the body of knowledge. The subject matter is presented in a comprehensive manner, and the references are appropriate for the topic being discussed.

ü  Line 17: 0. Introduction should be 1. Introduction

ü  Citations in the text should be made in order. In this manuscript, the authors cited from [1], [88], [89-92]….

ü  Figure 2: Please mention or explain it in the text.

ü  Figure 3(c): please add the dimensions of the drawing.

ü  Figure 4: Some numbers in the figure are unreadable. Please modify them.

ü  Line 60: what is “Google Academic”? Please confirm if it is “Google Scholar”?

ü  Tables: Please make Tables of some important references which could be the best step in the fields. Also, help readers ease understanding.

ü  Please translate references (in Chinese) into English. I am so sorry that I am not good at Chinese. They are [4, 6, 7, 22, 32, 40, 47, 53, 76, 81, 87].

ü  There are 106 symbols of [J] in the manuscript. What is it?

ü  References [62, 58, 37,103] are uppercase. Please lowercase as the journal template.

 2. Questions and suggestions:

 2.1 Your reference in this manuscript is 118, but you mention, "The references in this paper come from two resource libraries: CNKI and Google Academic. Using rubber, fatigue, and life as keywords to search, a total of about 120 articles published in recent 20 years were obtained.”. Please revise how many articles you have referred.

2.2 Why didn’t you choose Web of Science or Scopus Data instead of two resource libraries: CNKI and Google Scholar?

2.3 Please rewrite the abstract and the conclusions (new conclusions from already published works) to remark on significant stages of the field development. Besides, please identify the research gaps and explore potential areas in a particular field.

The reviewer highly appreciates your work. The reviewer hopes that his point of view could help the authors improve their work well.

Thank you.

Sincerely yours,

The Reviewer

Author Response

Dear Reviewers,

Hello, thank you for your suggestions. I will answer and modify your suggestions one by one.

ü Line 17: I have reordered the manuscript;

ü Citations: The references have been reordered;

ü  Figure 2: I added it in the manuscript;

U Figure 3 (c): the dimensions of the drawing has been added;

Figure 4: The drawing has been remade;

U Line 60: I made a mistake here. It's Google Scholar, not Google Academic;

U Tables: Sorry, I'm a little confused. I didn't think about what can be made into tables;

U references: References have been translated into English;

U [J]: Sorry, I didn't see this symbol in the text;

U: The case problem has been corrected.

2.Questions and suggestions:

2.1 I have revised the number to 119 in my manuscript.

2.2 The Chinese literature in CNKI is relatively complete, and the English literature in Google Scholar is relatively complete, so I chose these two resource libraries.

2.3 The summary and conclusion have been rewritten, please consult.

I look forward to your reply.

Reviewer 3 Report

Current manuscript entitled “Research Progress on Fatigue Life of Rubber Materials” by “Qiu et al” deliberated on the recent advancements on the rubber materials. Authors addressed the current research status of rubber fatigue and summarized in three aspects: research methods of rubber fatigue, factors affecting fatigue life and crack section, including the comparison between different rubber specimens, the summary of two research methods, and the effects of mechanical conditions, rubber composition and environmental factors on rubber fatigue. The section of rubber fatigue crack is expounded from macroscopic and microscopic perspectives. The topic is original and relevant in the field. The manuscript seems good, written well, and can be accepted after addressing the following comments.

1.     Please check thoroughly for the Grammatical errors

2.     Please discuss on the mechanical strength of the rubber materials.

3.     Some abbreviations are not properly defined, please check.

4.     In Figure incorporate the data for the year 2022

5.     Conclusion should be written as conclusions

6.     Mention the challenges that are currently facing with the “Fatigue Life of Rubber Materials”

7. Please improve the summary of two research methods.

Author Response

Dear Reviewer,

Hello, thank you for your suggestions. I will revise them one by one.
1. I have corrected the grammar errors I found and marked them in red.
2. I discussed the mechanical load borne by rubber in the manuscript, and I wonder if the mechanical strength still needs to be discussed.
3. I have added a definition after the abbreviation.
4. Sorry, I can't see the data of 2022 in the figure, so it hasn't been changed.
5. It has been revised, and I have merged the Expectation and Conclusions into one chapter.
6. Please check whether the current manuscript meets the requirements.
I look forward to your reply. Thank you.

Round 2

Reviewer 2 Report

Please read the attachment.

Thank you.

Author Response

Dear Reviewer,

Thank you for your letter. I will correct and answer your suggestion.
Line 68: It has been corrected.
Line 4: It has been corrected.
Figure 3(c): I have re-dimensioned, please check.
Tables: I made a chart, please see if it is suitable.
[J], [M], [D], [C]: I refer to the references cited by GB/T 7714 standard, so there are these symbols.
Line 518: Modified.
Line 519-524: I put this passage in Chapter 4. I think it's more appropriate here.
2. Questions:
2.1 Because the articles in two databases, Web of Science and Scopus Data, can also be searched in Google scholar, these two databases were not written, and I have now added them.
I marked the revised parts in the manuscript in blue font.
Kind regards

Round 3

Reviewer 2 Report

The authors did not understand my questions and suggestions from the previous two rounds of review.

In this round, I suggest that the authors carefully study the suggestions and questions of the reviewer. If there is any problem, disagree. The counter-argument indicates that the authors argue more persuasively.

Please read the attachment for more information.

Author Response

Dear reviewer,

All your suggestions and questions have been revised. The details are shown in the attachment.

Kind regards.
